# Fibrinolytic and antibiotic treatment of prosthetic vascular graft infections in a novel rat model

**Mikkel Illemann Johansen** [1,2]*, **Søren Jensen Rahbek** [3], **Søren Jensen-Fangel**[1], **Gabriel Antonio S. Minero**[2], **Louise Kruse Jensen**[4], **Ole Halfdan Larsen**[5], **Lise Tornvig Erikstrup**[6], **Anders Marthinsen Seefeldt**[1], **Lars Østergaard**[1], **Rikke Louise Meyer**[2,7], **Nis Pedersen Jørgensen**[1]

**1** Department of Infectious Diseases, Aarhus University Hospital, Aarhus N, Denmark, **2** Interdiciplinary Nanoscience Center (iNANO), Aarhus University, Aarhus C, Denmark, **3** Research Center for Emergency Medicine, Aarhus University Hospital, Aarhus N, Denmark, **4** Faculty of Health and Medical Science, Department of Veterinary and Animal Sciences, University of Copenhagen, Copenhagen, Denmark, **5** Department of Clinical Biochemistry, Aarhus University Hospital, Aarhus N, Denmark, **6** Department of Clinical Microbiology, Aarhus University Hospital, Aarhus N, Denmark, **7** Department of Biology, Aarhus University, Aarhus C, Denmark

* Mikkjh@rm.dk

**Data Availability Statement:** All relevant data are within the paper and its Supporting Information files.

## Abstract

### Objectives

We developed a rat model of prosthetic vascular graft infection to assess, whether the fibrinolytic tissue plasminogen activator (tPA) could increase the efficacy of antibiotic therapy.

### Materials and methods

Rats were implanted a polyethylene graft in the common carotid artery, pre-inoculated with approx. 6 log$^{10}$ colony forming units (CFU) of methicillin resistant *Staphylococcus aureus*. Ten days after surgery, rats were randomized to either: 0.9% NaCl (n = 8), vancomycin (n = 8), vancomycin + tPA (n = 8), vancomycin + rifampicin (n = 18) or vancomycin + rifampicin + tPA (n = 18). Treatment duration was seven days. Approximately 36 hours after the end of treatment, the rats were euthanized, and grafts and organs were harvested for CFU enumeration.

### Results

All animals in the control group had significantly higher CFU at the time of euthanization compared to bacterial load found on the grafts prior to inoculation (6.45 vs. 4.36 mean log$^{10}$ CFU/mL, p = 0.0011), and both the procedure and infection were well tolerated.

Vancomycin and rifampicin treatment were superior to monotherapy with vancomycin, as it lead to a marked decrease in median bacterial load on the grafts (3.50 vs. 6.56 log$^{10}$ CFU/mL, p = 0.0016). The addition of tPA to vancomycin and rifampicin combination treatment did not show a further decrease in bacterial load (4.078 vs. 3.50 log$^{10}$ CFU/mL, p = 0.26). The cure rate was 16% in the vancomycin + rifampicin group vs. 37.5% cure rate in the

**Funding:** This work was supported by a PhD research grant from the Graduate School of Health, Aarhus University (MIJ), by Aarhus Universitets Forskningsfond (https://auff.au.dk/en/) (MIJ) and by Knud og Edith Eriksen Mindefond (Grant number 62786) (NPJ)(https://www.keemindefond.dk/). The funders did not play any role in the study design, data collection and analysis, decision to publish or preparation of the manuscript.

**Competing interests:** The authors have declared that no competing interests exist.

vancomycin + rifampicin + tPA group. Whilst interesting, this trend was not significant at our sample size (p = 0.24).

## Conclusion

We developed the first functional model of an arterial prosthetic vascular graft infection in rats. Antibiotic combination therapy with vancomycin and rifampicin was superior to vancomycin monotherapy, and the addition of tPA did not significantly reduce bacterial load, nor significantly increase cure rate.

## Introduction

Prosthetic vascular grafts are vulnerable to bacterial infections, and implant-associated infection is a serious complication, associated with both high morbidity and mortality [1]. The infection rate is approximately 3%, depending on the type of implant and anatomical location, a risk that persists for the remainder of the patient's life [2]. Implant-associated infections are characterized by biofilm formation—a community of bacteria that exhibits an altered phenotype characterized in part by recalcitrance to antibiotic therapy [3,4]. Staphylococci are among the most frequent pathogens isolated from implant-associated infections [5].

The interaction between mammalian extracellular matrix molecules and *Staphylococcus aureus* is a key aspect of *S. aureus* biofilm pathology [6–9]. Of particular interest is the ability of *S. aureus* to evade the host organisms' immune system by formation of a fibrin scaffold through enzymatic conversion of fibrinogen to fibrin [10–12].

This interaction should be considered a target for new treatment options, as targeting the fibrin scaffold could potentially leave the bacteria vulnerable to antibiotics and the host's immune response [11]. We have previously shown *in vitro*, that enzymatic dispersal of the bacterial biofilm with streptokinase increases the efficacy of antibiotic treatment against *S. aureus* biofilm infections [13]. However, tissue plasminogen activator (tPA) has replaced streptokinase for stroke treatment, as streptokinase treatment has an increased risk of hemorrhage. Further tPA has a more specific effect on fibrin bound plasminogen, and hence makes an attractive drug for repurposing for treating prosthetic vascular graft infection (PVGI) [14]. Although this concept has been tested *in vivo* in intravascular catheter (IVC) infections, it has not been tested in a PVGI model [15]. To test this, we developed a model in which a graft, inoculated with Methicillin-resistant *Staphylococcus aureus* (MRSA), was implanted in the artery of rats. This would In our opinion better reflect the clinical situation of a vascular graft infection.

We then hypothesized that vancomycin, in combination with rifampicin, would be superior to monotherapy with vancomycin for the treatment of acute PVGI. Finally, we hypothesized that addition of fibrinolytic treatment using tPA would improve treatment outcome.

## Materials and methods

### Ethical statement

This study was approved by the Danish Animal Experiments Inspectorate under permission 2017-15-0201-01153 and 2022-15-0201-01132 and was carried out in accordance with the recommendations in the Guide for the Care and Use of Laboratory Animals of the National Research Council and under the supervision of the Institute of Biomedicine, Aarhus

University's veterinarians. To comply with the 3Rs, we chose to power our study to test the best antibiotic treatment ± tPA. This would allow us to test our hypothesis if tPA would increase efficacy of the treatment we find most clinically relevant. Sample size calculation was based on existing models, with an assumption of <50% cure rate for vancomycin and rifampicin treated animals and a >90% cure rate using tPA and antibiotic combination treatment with an alpha value of 0.05, a beta value of 0.20 and a power of 80% resulted in a sample size of 18 animals pr. treatment group.

## Study animals

A total of 76 adult male Sprague-Dawley rats (Janvier Labs, Le Genest-Saint-Isle, France) (300-350g) were included in the study. Rats were housed at the Institute of Biomedicine, Aarhus University, at standard room temperature, humidity and 12 h day/night cycle with free access to food and water. Animals had an acclimatization period of one week after arrival.

## Bacterial strain preparation

For all experiments we used the isolate USA300 FPR3757 (ATCC® BAA-1556), a clinical MRSA isolate from an abscess infection [16]. Colonies from a 48-h old culture, grown on brain heart infusion (BHI) agar plates, were incubated overnight in BHI media for 18 h at 37˚C. The overnight culture was then centrifuged for 5 min at 503 g, and washed in phosphate buffered saline (PBS). This was repeated three times. The culture was then diluted in a modified BHI media (supplemented with 2.1 mM $CaCl_2$ and 0.4 mM $MgCl_2$) to $OD_{600}$ = 0.1 ($5 \times 10^7$ CFU/mL).

## *In vitro* biofilm dispersal assay

Biofilm were grown using the assay by Jørgensen et al [13]. Peg-lid (NUNC-TSP 445497, Thermo Fisher) were preconditioned in BHI enriched with 10% fresh human plasma for 30 min at 37˚C and then transferred to a 96-well (NUNC 161093, Thermo Fisher) with OD-adjusted overnight culture ($OD_{600}$ = 0.1) and incubated for 2h at 37˚C. The peg-lid were then transferred back to the pre-conditioning plate and incubated for 24h at 37˚C. The peg-lids were treated in 96-well plates with tPA 2500 U/mL, ±10% plasma or streptokinase 500 U/mL (streptokinase from β-hemolytic *Streptococcus*, Sigma-Aldrich) ±10% plasma and incubated for 30 min at 37˚C. Following treatment, the peg-lid was air-dried for 30 min at room temperature, then transferred to a 96-well plate containing Gram's crystal violet (Sigma-Aldrich) solution and incubated for 10 min at room temperature. The peg-lid was washed twice in demineralized water and then air-dried for 30 min at room temperature. The Gram stains on the peg-lid were extracted with 33% acetic acid and the absorbance of the extracted stain was read with a plate reader at 590 nm.

## Antibiotic susceptibility testing using minimum biofilm eradication concentration

Biofilms were prepared as described above, but peg-lids were preconditioned with plasma for 60 min instead of 30 min, and rotational shaking 50 rpm was added to the incubation step. After incubation for 24h, the peg-lids were transferred to new 96-well plates containing fresh BHI media and 10% human plasma and incubated for 24h at 37˚C, 50 rpm. The peg-lids were then transferred to two 96-well plates containing vancomycin in two fold dilutions and rifampicin at a fixed concentration (10 mg/L), one plate was added tPA (2500 U/mL) to all wells and then incubated for 24h at 37˚C, 50 rpm. Peglids were then washed twice in fresh BHI media

for 1 min, and transferred to a 96-well plate containing fresh BHI media and sonicated in a ultrasonication bath (USCS1700 T, VWR) at 45 kHz and 110 W for 10 min, followed by incubation for 24h at 37˚C. The minimum biofilm eradication concentration (MBEC) was determined by measuring $OD_{600}$ on a plate reader.

## Inoculation of vascular grafts

Polyethylene (Intramedic® Polyethylene Tubing PE 50) with a diameter of 0.965 mm (outer) and 0.53 mm (inner) was used as a vascular conduit. The PE tubing was filled with heparinized rat plasma and incubated for 2 h at 37˚C to precondition the grafts with extracellular matrix molecules. An overnight culture of MRSA was prepared, as described in the bacterial strain preparation. The overnight culture was then supplemented with 10% heparinized rat plasma, and filled into syringes. The syringes were connected to the PE tubing and then connected to a syringe pump (Harvard Apparatus, PHD ULTRA™ Syringe Pump) and the culture was run through at 200 µL/min for 2 h at 37˚C. The tubing was cut in to 0.8 mm tubes, and stored overnight in PBS-soaked gauze at 4˚C.

## Prosthetic vascular graft infection model

Following acclimatization, rats were given a cefuroxime prophylaxis 30 min before surgery (50 mg/kg, B. Braun) and anesthetized with isoflurane (5% induction, 2,5% maintenance, Zoetis) and atmospheric air 1,5 L/min. The throat was shaved (Fig 1A), disinfected with 70% ethanol and, following an initial incision, the carotid artery was exposed by blunt dissection and separated from the surrounding tissue (Fig 1B). As soon as the artery was clamped, a stop clock was started, and blood flow was stopped for approximately 10 min (Fig 1C). A small incision was then made in the artery (Fig 1D), and the graft was placed within the artery fixated with two ligatures to prevent dislocation (Fig 1E). The ligature on the carotid artery was then removed, and reestablishment of flow through the graft and artery was subsequently observed with the naked eye in all animals (Fig 1F). During the preliminary studies with uninfected animals, we verified flow in the grafts with ultrasound (GE Vivid S6, GE Healthcare Denmark A/ S). This was not possible during our later studies, as ultrasound was not available at the lab classified for work with MRSA. The rats were then returned to their cages once the incision was closed with sutures ("Monosyn, 5/0", B. Braun) and the surgical site was disinfected. (Fig 1G). Buprenophine (Temgesic®, Indivior) was administered in their water supply the first four days (0.05 mg/mL), and they were weighed every four days.

## Medical interventions

Ten days after initiation of infection, the rats commenced therapy. Animals were treated with antibiotics, reflecting best clinical practice [17], and tissue plasminogen activator (tPA) was chosen over other previously tested fibrinolytics, as this is the drug of choice for thrombotic events in the clinical setting [18]. The tPA dosage was chosen in order to correspond to a human dose for treating ischemic stroke [19]. Animals were divided in to five treatment groups: 1) 0.9% NaCl, (n = 8) 2) vancomycin 50 mg/kg (Fresenius Kabi, Bad Homburg, Germany) (n = 8) 3) vancomycin + tPA 0.9 mg/kg ("Actilyse", Boehringer Ingelheim,) (n = 8) 4) vancomycin + rifampicin 25 mg/kg ("Eremfat", Riemser) (n = 18) 5) vancomycin + rifampicin + tPA (n = 18). The animals were randomly placed in their cages by the staff at the animal facilities, three rats in each. Each cage was then randomized into one of the five treatment groups by letter randomisation. Antibiotics were injected intraperitonially every 12 h [20]. Doses of tPA were administered intravenously in the tail once daily, 30 min after antibiotic administration.

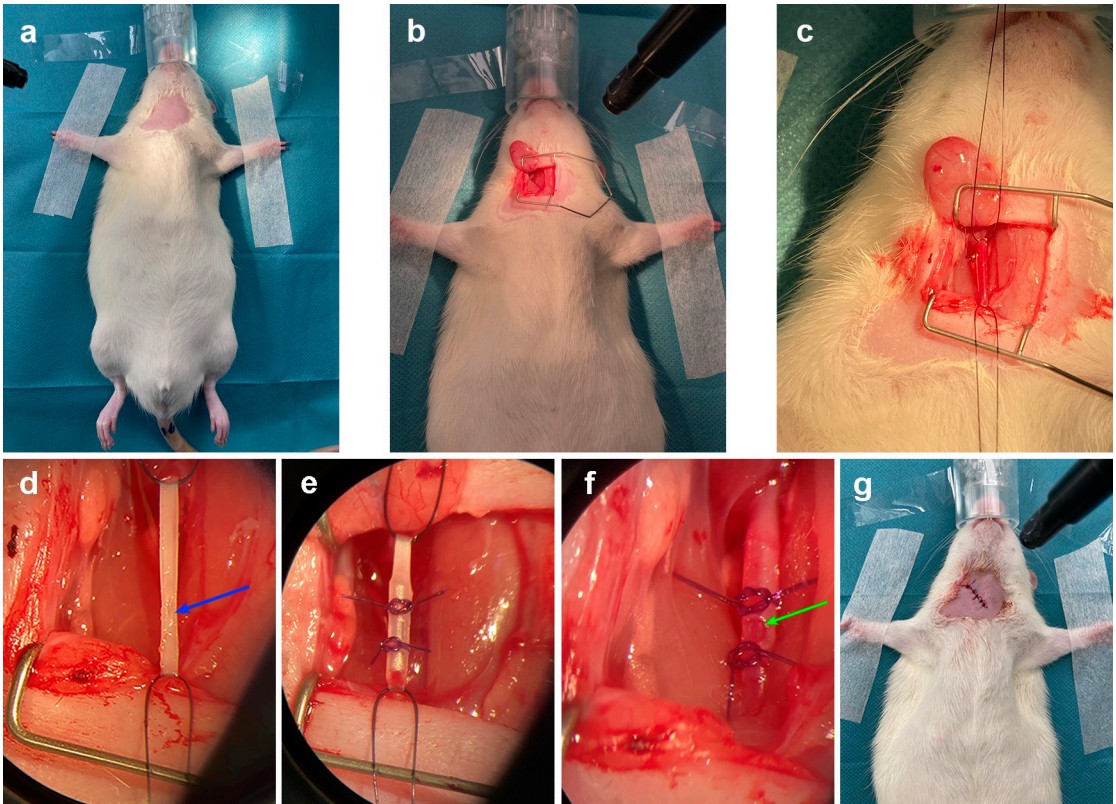

**Fig 1. Surgical procedure of rat PVGI model.** (**a**) Anesthetized rat with shaved and exposed chest prior to surgery. (**b**) Incision with view to the space created between the right sternocleidomastoid and the omohyoid muscle. (**c**) Common carotid artery exposed and ligated. (**d**) Blue arrow shows exposed common carotid artery, prior to insertion of graft. (**e**) Graft inserted in common carotid artery and secured with ligatures. (**f**) Green arrow shows graft with blood flow inserted in to common carotid artery. (**g**) Rat post-surgery, with wound closed with sutures.

Rats were treated for 7 days, followed by 36 h of observation after the last antibiotic administration to ensure washout of antibiotics prior to sampling for CFU measurements. Animals were euthanized by cervical dislocation under isoflurane anaesthesia, grafts were removed, stored in 1 mL PBS and immediately subjected to analysis for quantification, and samples of organs (liver, spleen and kidney) were harvested and stored at -80°C.

## Quantitative analysis of bacterial load from grafts and organs

Explanted grafts were submerged in 1 mL PBS and vortexed for 30 s in a Vortex Genie (Scientific Industries, Inc.) and subsequently sonicated in an ultrasonication bath at 45 kHz and 110 W for 5 min (USCS1700 T, VWR). The tubes were vortexed for 30 s and the bacterial suspension of each graft was serially diluted in triplicates, followed by plating onto BHI agar plates and incubated for 48 h at 37°C. Enumeration of CFU was done on the agar plates, and bacterial load was calculated. Organ samples were homogenized at 2 x 20 seconds 5000 rpm (Precellys 24 tissue homogenizer, Bertin Technologies), stored in wet ice for 5 min and run for one more session. CFU enumeration was done on the homogenate as described above. Following CFU measurements, the remaining sonicate and the graft from each animal were added to 20 mL of fresh BHI media and grown overnight at 37°C. This was done to increase the bacterial detection limit and have previously been done by both Jørgensen et al and Metsemakers et al

[21,22]. All animals plotted as 0 CFU/mL had no growth on the agar plate or in the overnight culture of the graft with remaining sonicate.

## Bacterial species identification and rifampicin resistance measurements

Following CFU enumeration, bacterial colonies from BHI agar plates were subcultured on 5% blood agar plates (Statens Serum Institute (SSI) Diagnostica) and incubated for 18–24 h at 35–37˚C. After incubation, colonies from 5% blood agar plates were confirmed to be *S. aureus* with matrix-assisted laser desorption/ionization time-of-flight mass spectrometer (MALDI-TOF) (Bruker Microflex, LT/SH system; Bruker Daltonik GMbH). To confirm rifampicin susceptibility, minimum inhibitory concentration (MIC) determination was performed by gradient test (Etest strips, bioMérieux). For the preparation of inoculum, inoculation, and incubation, we followed the guidelines recommended by the European Committee on Antimicrobial Susceptibility Testing (EUCAST) (http://www.eucast.org). All MALDI-TOF and rifampicin susceptibility tests were performed at the reference laboratory at the Department of Clinical Microbiology at Aarhus University Hospital by trained staff.

## Confocal laser scanning microscopy of infected prosthetic vascular grafts

A total of nine rats were used for visualisation of fibrin and biofilm on the graft. Following the protocol for the PVGI model described above, animals were divided into three groups: 1) 10 days infection, no treatment (n = 3), 2) 19 days infection, no treatment (n = 3), 3) 19 days infection, seven days treatment with vancomycin+rifampicin+tPA (n = 3). At the end of infection period, animals were euthanized and grafts were removed for immediate analysis. The vascular grafts were sliced into round as well as oblong slices and treated (1) in a blocking solution of 3% bovine serum albumin (BSA) in 1x PBS for 3 min, followed by (2) immunolabelling of the slices in the blocking solution containing 10 mg/L anti-fibrin antibody (59D8, Atto488 conjugated rabbit IgG) (Absolute Antibody) for 75 min, followed by (3) washing the slices in the blocking solution and (4) staining the bacterial and immune cells in 10 μM SYTO41 (intracellular DNA binding stain, Thermofisher Scientific) solution in 100 mM NaCl. The slices were then visualized by confocal laser scanning microscopy using Zeiss 20x objective and the following settings: 488 nm laser (4.0%, gain 1200) as track 1 and 405 nm laser (2.0%, gain 900) as track 2.

## Histological analysis

6 rats were used for histological and immunohistochemical analysis. Rats were operated following the PVGI protocol described above and divided in to two groups: 1) sterile implant, euthanized day 19 post-surgery (n = 3), 2) MRSA infected implant, euthanized day 19 post-surgery (n = 3). Following euthanasia, the left and right arteries, including surrounding soft tissue, were dissected free in blocks of 5 x 5 mm. For the right artery, the vascular prosthesis was not removed and thus fixated and processed for histology *in situ*. All tissue blocks were fixated in 10% natural buffered formalin for 2 weeks. Afterwards, tissue blocks were cross-sectionized and processed through graded concentrations of alcohol and embedded in paraffin wax. Sections of 2–3 μm were cut from both the right and left side and Haematoxylin and Eosin (HE) staining was used for evaluation of patho-morphology. In addition, sections from the right artery were also cut and stained with PTAH and Masson Trichrome to visualize fibrin and collagen, respectively. Neutrophil infiltration and fibroplasia were semi-quantitatively scored as absent, mild, moderate, or massive. Histology was performed by a blinded pathologist, unaware of group allocation and study design.

## Immunohistochemistry

Sections of 2–3 μm were cut from paraffin embedded tissue blocks of the right artery and used for immunohistochemistry with a primary antibody against Staphylococci. The protocol has recently been published (Jensen LK 2023, APMIS). Briefly, no antigen retrieval procedure was carried out and sections were subjected to blocking of endogenous peroxidases in 0.6% $H_2O_2$ for 15–20 minutes. UltraVision Protein Block (AH Diagnostics, Tilst, Denmark) for five minutes was used to prevent additional unspecific staining. Sections were incubated overnight at 4˚C with the primary antibody (Polyclonal mouse anti-goat staphylococci antibody, catalogue PA1-7246, ThermoFisher, Roskilde, Denmark) diluted 1:38400 in 1% BSA/TBS. Immunostaining was carried out using the UltraVision indirect Horse Radish Peroxidase (HRP) polymer-amplification technique. Following incubation with HRP-polymer (LabVision, Värmdö, Sweden) the sections were incubated with a red chromogen solution (AEC from LabVision). All stains were counterstained with Mayers haematoxylin. Negative staining controls were run on parallel sections and included substitution of the primary antibody with 1% BSA and a non-sense antibody (Rabbit immunoglobulin fraction, catalogue X0903, DAKO A/S, Glostrup, Denmark) of the same concentration as the primary reagent, respectively. A porcine lung experimentally infected with *S. aureus* was used as positive control. All tissue slides were evaluated under high power (400 x magnification) and the presence and localization of red positive bacteria was registered. IHC was performed by a blinded pathologist, unaware of group allocation and study design.

## Statistical analysis

The distribution of data was assessed using the Shapiro-Wilk test. Differences in mean biofilm biomass were assessed via unpaired Student's t-test and presented as mean ± SD. Differences in weight changes and log CFU/mL were presented as mean ± SD for normally distributed data, and assessed via two-tailed, unpaired Mann-Whitney test for non-parametric data, presented as median with interquartile range (IQR). This was followed by a post hoc analysis using the Holm-Bonferroni test. Differences in cure rate were assessed via Fisher's exact test. All statistical calculations and figures were made with GraphPad Prism 9.0.0 (Graphpad Software).

# Results

## tPA and streptokinase were able to partly disperse biofilms grown *in vitro*

tPA and streptokinase were able to partially disperse a 24h old *S. aureus* biofilm. 30 min treatment with tPA lead to a 1.6 fold reduction in biofilm biomass from 12.41 to 7.72 (p = 0.038) and streptokinase treatment lead to a 1.7 fold reduction in biofilm biomass from 12.41 to 7.16 (p = 0.024). The addition of plasma to the treatment step did not lead to a further decrease in biofilm biomass for streptokinase although (p = 0.0086). However, adding plasma to the treatment step diminished the effect of tPA leading to a 1.3-fold reduction in biofilm biomass from 12.41 to 9.20 (p = 0.091).

## tPA does not increase biofilm antibiotic susceptibility *in vitro*

tPA was not able to decrease the MBEC value for vancomycin, in fact there was a 2-fold increase in vancomycin MBEC when tPA was added to treatment (Table 1). Similarly, an increase in vancomycin MBEC was observed when combined with rifampicin and tPA. Regardless of adding tPA or not, vancomycin MBEC was dramatically reduced when adding rifampicin.

**Table 1. Minimum biofilm eradication concentration for MRSA FPR3757.**

| Antibiotic | MBEC (mg/mL) | |
|---|---|---|
| | -tPA | +tPA |
| **Vancomycin** | 256 | 512 |
| **Vancomycin + Rifampicin (10 mg/L)** | 2 | 8 |

Table 1: Impact of tPA on MBEC values for MRSA FPR3757 biofilms treated with vancomycin or vancomycin +rifampicin. Rifampicin was kept at a fixed concentration of 10 mg/L.

## Animal welfare initially decreased following surgery, but initial weight loss was regained by the 8th postoperative day

A total of 58 of 61 animals completed the treatment study and were included in the final dataset. Three animals were excluded: one died whilst receiving a tPA injection, and subsequent autopsy failed to establish cause of death but no signs of haemorrhage were evident; one died immediately following the initial surgery, and one was excluded due to the absence of growth on agar plates, despite having macroscopic pus in the graft.

Clinically, the animals fared well. There was an initial weight loss following surgery in all groups. By day 12 post-surgery, all animals had regained the lost weight and weight increased throughout the treatment period, with an average weight increase of 10.1% (6.3%-17%) (Fig 2). Over the 19 days of infection, the mean weight increase was 17% ± 5.4 for the NaCl group (n = 8), 12.7% ± 2.8 for the vancomycin group (n = 8), 7.3%±2.9 for the vancomycin+tPA group(n = 8), 6.3%±2.9 for the vancomycin+rifampicin group (n = 18), and 7.0%±3.8 for the vancomycin+rifampicin+tPA group (n = 16).

## Bacteria from grafts were *S. aureus* monomicrobial infection and systemic dissemination of infection was present in all groups, regardless of intervention

Mean bacterial load on the grafts after 19 days of infection in NaCl group was significantly higher than the bacterial load measured on grafts prior to implantation (an increase of 2.310 ±0.49 $\log^{10}$ CFU/mL, mean±SD, p = 0.0011 Students t-test). MALDI-TOF analysis confirmed that all samples grew monomicrobial *S. aureus*. Using the EUCAST clinical breakpoint for rifampicin (susceptible, MIC ≤0,06 mg/L), all isolates were susceptible to rifampicin with an MIC below 0.06 mg/L (Table 2). Regardless of intervention, the percentage of infected organs did not differ (Fig 3A) nor was there a difference between the number of animals with disseminated infection, i.e., percentage of animals from which viable bacteria was grown from either liver, kidney or spleen in addition to the graft (Fig 3B).

## tPA does not affect fibrin in biofilms on infected prosthetic vascular grafts

Confocal laser scanning microscopy was used to investigate if the vascular grafts contained a biofilm with fibrin. After 10 and 19 days of infection, clusters of tiny cells covered the surface of the implant, which was interpreted as being bacterial biofilms, although we were above the resolution (20x objective) to see the detailed outline of bacteria (S1 Fig and Fig 4). Fibrin was primarily found at the implant surface and interspersed between the bacteria, thus it was incorporated into the biofilm. Moreover there was an abundance of larger cells presumably murine immune cells clustered around the smaller cells and fibrin.

Quantification of the total amount of fibrin from the graft after 19 days of infection revealed that there was no significant difference in amount of fibrin on the grafts between animals

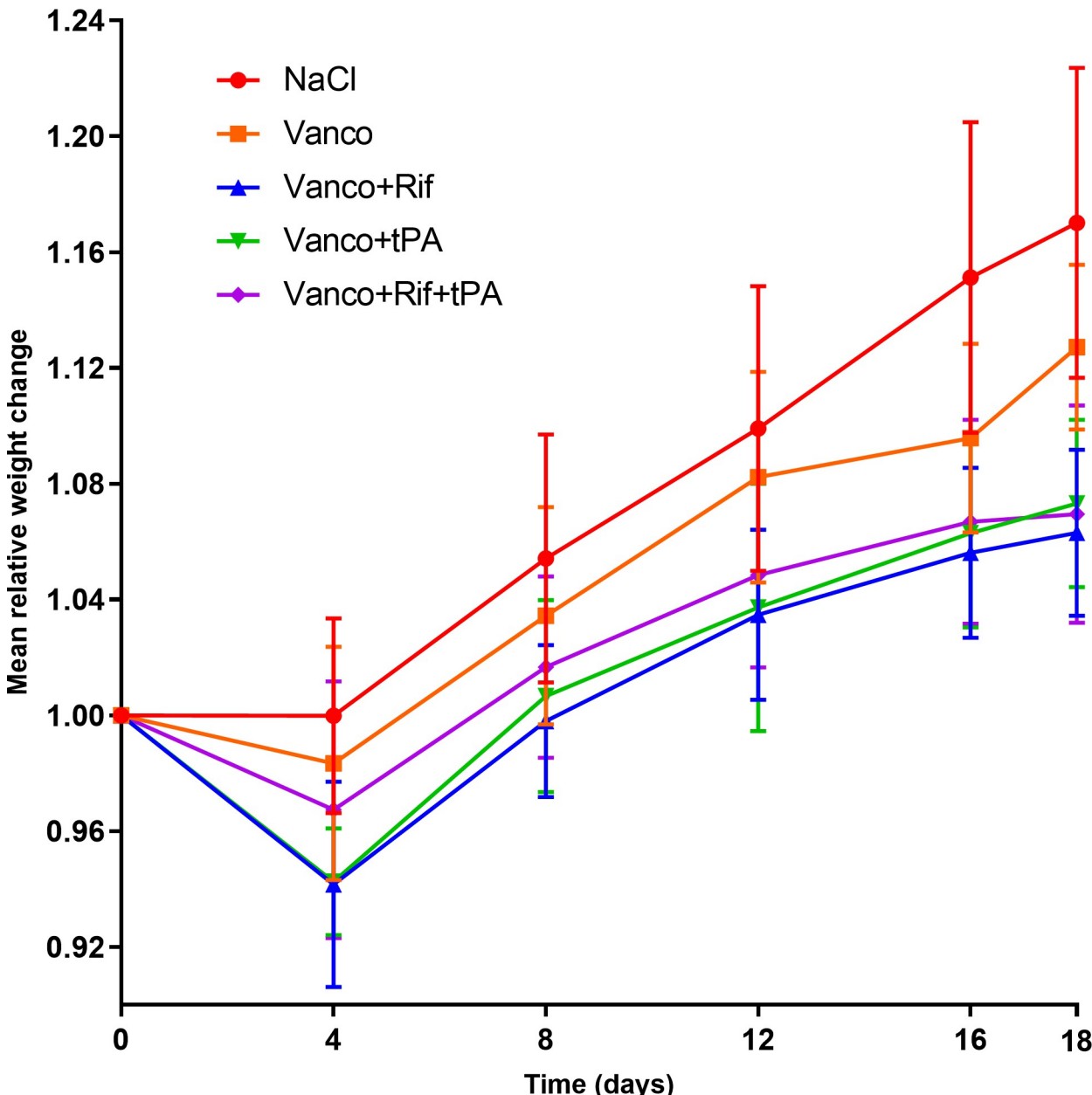

**Fig 2. Mean relative weight changes post-infection compared to starting weight.** NaCl (n = 8), Vanco (n = 8), Vanco+rif (n = 18), Vanco+tPA (n = 8), Vanco+Rif+tPA (n = 16). Vanco: Vancomycin, Rif: Rifampicin, tPA: tissue plasminogen activator. Symbols are mean values whereas error bars show one standard deviation.

treated with NaCl and animals treated with vancomycin+rifampicin and tPA (Fig 4C). However, one of the explanted grafts from the tPA group did not contain any fibrin.

## Profound histological changes evident after 19 days of infection

Briefly, the rat carotid artery wall contains four layers of elastic lamina which appear as pink lines in HE sections (Fig 5A). The elastic laminas are separated by smooth muscle cells. The area between the outermost and the innermost lamina is the tunica media (Fig 5A). The area

**Table 2. Qualitative results on bacterial growth from explanted grafts.**

| Treatment (n = 58) | + Growth (n = 49) | - Growth (n = 9) | Isolates susceptible to rifampicin, n (%) (MIC <0.06 mg/L) | Cure rate (%) |
|---|---|---|---|---|
| Vancomycin + rifampicin (n = 18) | 15 | 3 | 100 | 16 |
| Vancomycin + rifampicin + tPA (n = 16) | 10 | 6 | 100 | 37.5 |
| Vancomycin (n = 8) | 8 | 0 | N/A | 0 |
| Vancomycin+tPA (n = 8) | 8 | 0 | N/A | 0 |
| NaCl (n = 8) | 8 | 0 | N/A | 0 |

Table 2: Comparison of most effective treatments against PVGI, and amount of isolates showing rifampicin resistance. Results are shown as number of agar plates with or without visible growth.

towards the vessel lumen from the innermost lamina is the tunica intima, a monolayer of endothelial cells in intact vessels. Outside tunica media is the tunica adventitia located (Fig 5A).

**Group 1.** Animals in group 1 (sterile implant) were euthanized day 19 post surgery. In all animals, the graft lumen showed varying degree of fibrin exudation intermingled with erythrocytes. All animals had a broken tunica interna with neointimal hyperplasia including collagen deposition by fibroblasts (Fig 5B). Neointimal hyperplasia occurs because vascular injury activates smooth muscle cells in tunica media to induce neointima formation by cell proliferation and migration into tunica intima (Welt, F. G. & Rogers 2002). Tunica media was found atrophic in all animals, i.e., loss of smooth muscle cells. Commonly, elastic fibers of tunica media were broken and in two cases they were completely lost in focal areas, and the vessel wall was completely replaced by fibroplasia. Mineralization of tunica media was seen in two animals. Tunica adventitia was found fibrotic and with macrophage infiltration (Fig 5B). Peri-adventitial fibrosis, macrophage infiltration and dilated fibrotic lymph vessels was seen in all animals while two animals presented with peri-adventitial mineralization. Peri-adventitial granulomatous inflammation in relation to foreign bodies made by sutures were occasionally seen. Neutrophil infiltration was not observed in any tunica layer or in surrounding tissues.

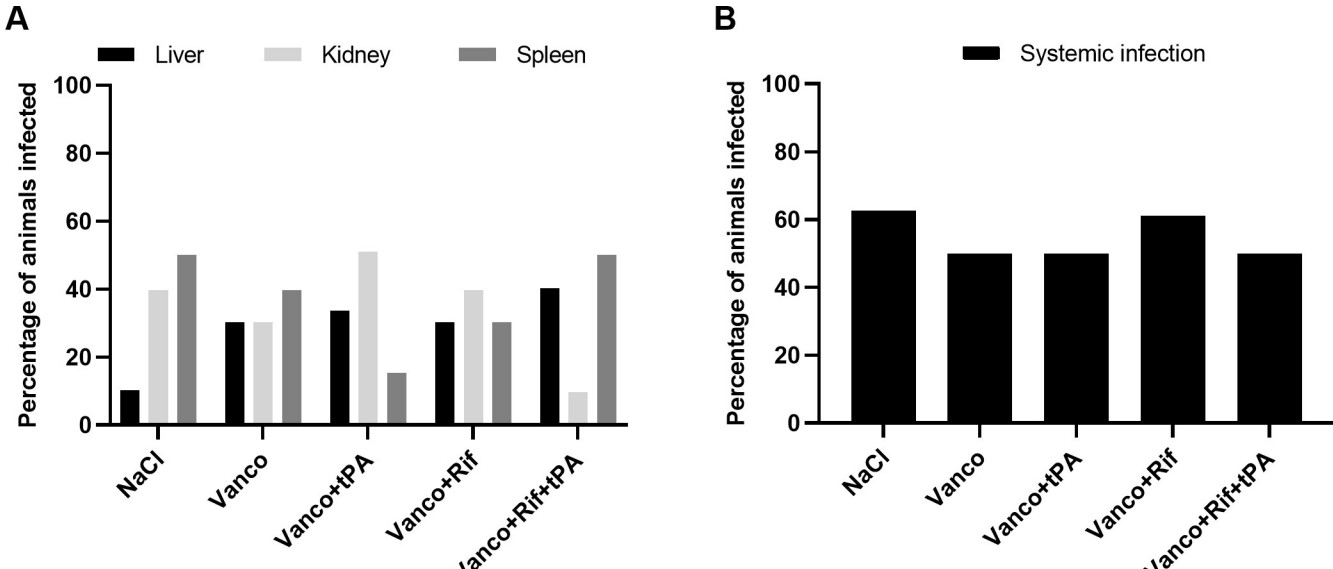

**Fig 3. Dissemination of infection.** (**A**) Percentage of animals in each treatment group with infections in the liver, kidney or spleen. The animals could have infection in more than one organ. (**B**): Percentage of animals in each treatment group with systemic infection (infection in either liver, kidney or spleen). NaCl (n = 8), Vanco (n = 8), Vanco+tPA (n = 8), Vanco+Rif (n = 18), Vanco+Rif+tPA (n = 16). Vanco = Vancomycin, Rif = Rifampicin.

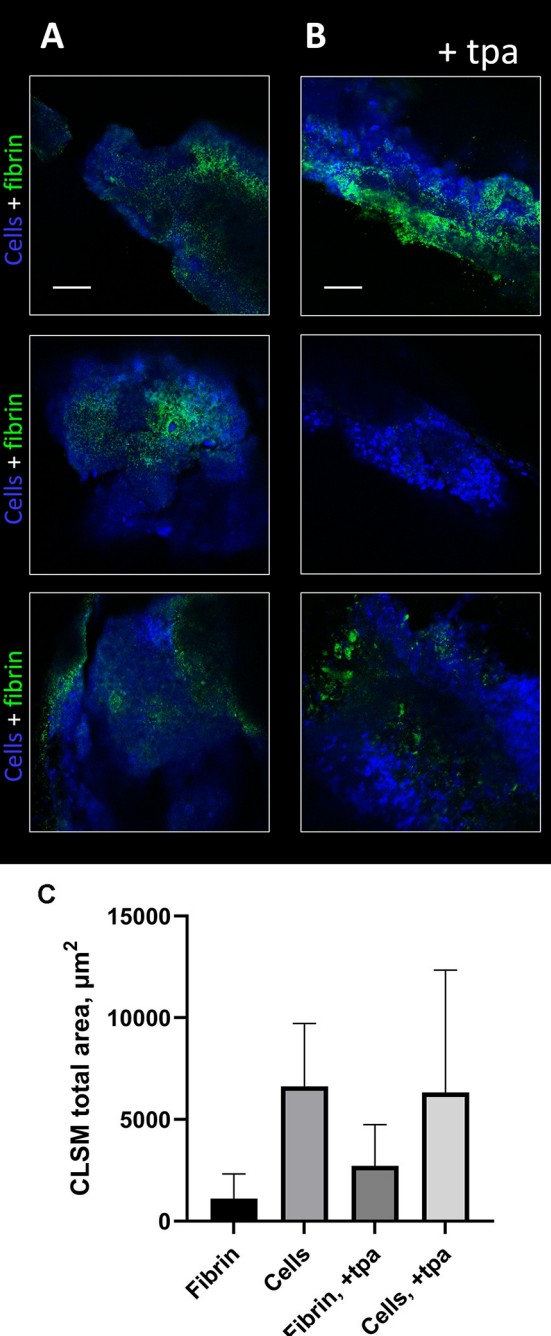

**Fig 4. Tissue plasminogen activator (tpa) does not affect fibrin in the in vivo biofilm from rat prosthetic vascular graft infections.** CLSM images of the MRSA biofilm formed in prosthetic vascular grafts in rats after 19 days of infection (A) treated with NaCl and (B) treated with tPA. Cells (bacterial and murine) are shown in blue (SYTO41) and fibrin is shown in green (anti-fibrin antibody, 59D8, Atto488 conjugated rabbit IgG). The 2D images are shown as two-channel. Scale bar = 50 μm. (C) Quantification of 2D CLSM images in Daime based on three biological replicates (total n = 18).

**Group 2.** Animals in group 2 (infected implants) were euthanized 19 days post-surgery. In all animals, the graft lumen showed varying degree of fibrin exudation intermingled with erythrocytes and massive amounts of neutrophil granulocytes (Fig 5C). All animals had a

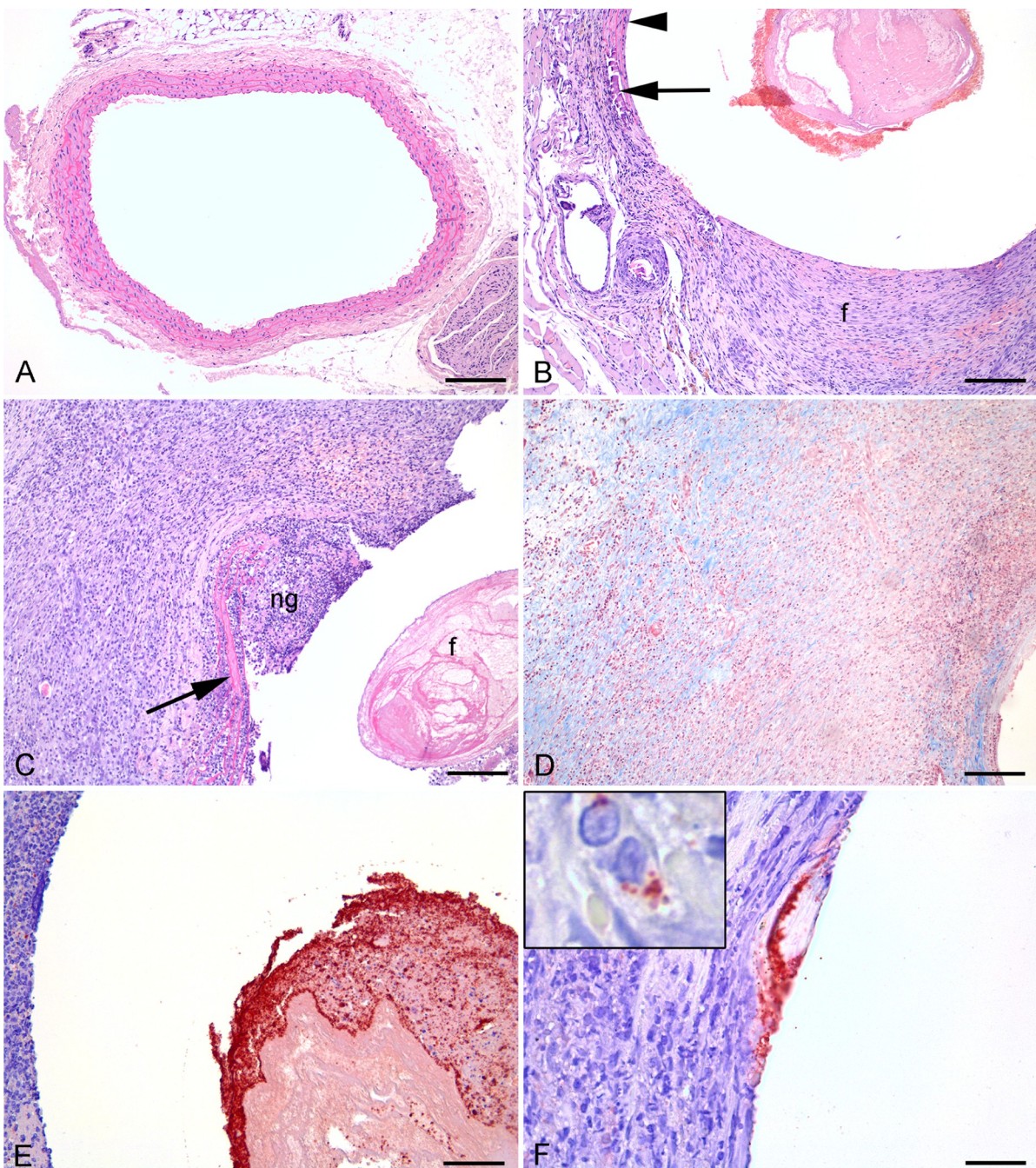

**Fig 5. Histological and immunohistochemic images of infected and sterile graft, artery and sorrounding tissue from rats 19 days after insertion.** A: Normal left a. carotis (HE), bar = 250 μm. B: Right a. carotis 19 days after insertion of a vascular graft. Neointima (ni) formation, tunica media atrophy (arrow) and fibrosis of tunica adventitia (HE), bar = 100 μm. C-F: Right a. carotis 19 days after insertion of a vascular graft and inoculation with *Staphylococcus aureus*. C: Pink fibrin (f) is present within graft lumen. Neutrophil granulocyte (ng) infiltration, broken elastic bands (arrow) and surrounding macrophages and collagen producing fibroblasts (HE), bar = 200 μm. D: Masson trichrome staining to identify collagen in blue, bar = 200 μm. E: Red *S. aureus* positive bacteria located on fibrin within graft lumen (IHC), bar = 100 μm. F: Red *S. aureus* positive bacteria located on the vessel surface towards the graft. Insert shows red positive bacteria within a macrophage (IHC), bar = 80 μm.

broken tunica interna and loss of normal tunica media tissue architecture (Fig 5C). The vascular grafts were surrounded by a massive suppuration of neutrophil granulocytes which again was surrounded by massive fibroplasia and macrophage and neutrophil granulocyte infiltration (Fig 5C). Inside this area remnants of tunica media could be found, i.e., broken and curled elastic fibers with loss of smooth muscle cells (Fig 5C). The lesions continued into peri-adventitial fibrosis with massive macrophage and neutrophil granulocyte infiltration and dilated fibrotic lymph vessels (Fig 5D). One animal had a peri-adventitial abscess, i.e., bacteria surrounded by neutrophil granulocytes and with fibrotic encapsulation. Peri-adventitial granulomatous inflammation in relation to foreign bodies made by sutures were seen in one animal.

## Immunohistochemistry reveal abundance of bacteria at day 19 post inoculation with intracellular location in neutrophils and macrophages

Positive and negative staining controls performed as expected, i.e., for the negative control application of 1% BSA and nonsense antibodies, respectively, resulted in sections without positive immunolabelling. For the positive control coccoid bacteria located centrally within the lung abscesses was found highly reactive, i.e., colored in red. Positive bacteria could not be identified in any animals of Group 1. In contrast, positive bacteria were identified in all Group 2 animals. Bacteria were identified in the graft lumen of all animals and in small colonies adjacent to the graft (Fig 5E and 5F). Intracellular bacteria were identified in neutrophil granulocytes and especially macrophages throughout the lesions (Fig 5F), i.e., also in peri-adventitial macrophages several mm from the graft. Bacteria of the peri-adventitial abscess was also found positive.

## Rifampicin combination improves vancomycin treatment against acute PVGI but the addition of tPA does not increase efficacy

Monotherapy with vancomycin was ineffective, as it did not result in a significant reduction in median bacterial load on the implants, when compared to the NaCl control group (6.91 $\log^{10}$ CFU/mL, IQR 5.10–7.50 for vancomycin vs. 6.45 $\log^{10}$ CFU/mL, IQR 5.10–7.64 for NaCl, p = 0.88) (Fig 6). Adding rifampicin to vancomycin resulted in a median $\log^{10}$ CFU/mL of 3.50 (IQR 2.09–5.28) (n = 18) which was significantly lower than monotherapy vancomycin (p<0.0001, n = 8) and NaCl (p<0.0001, n = 8). The addition of tPA to antibiotic combination therapy did not decrease the bacterial load further (4.078 $\log^{10}$ CFU/mL, IQR 0.00–4.41, n = 16) when compared to vancomycin+rifampicin treatment (p = 0.26). Finally, the addition of tPA to vancomycin + rifampicin did not significantly improve the overall cure rate of the infection on the vascular grafts, as the cure rate in the vancomycin + rifampicin group was 16% vs. 37.5% cure rate in the vancomycin + rifampicin + tPA group (p = 0.25) (Table 2).

## Discussion

To our knowledge, this is the first rat model of a PVGI, in which the implant is located in an artery. This model closely mimics the environmental conditions of biofilm infections in vascular grafts, taking into consideration both interaction between host molecules and the specific implant material used for prosthetic vascular grafts, blood flow in the arterial circulation and the host immune response to an intravascular infection following major surgery. Previous models have primarily focused on IVC and subcutaneous implant models with vascular graft material [15,23–26]. Only a single study has developed a mouse model of PVGI, but a murine model is not suited for testing fibrinolytic drugs, as mice generate substantially less plasminogen than rats [27,28].

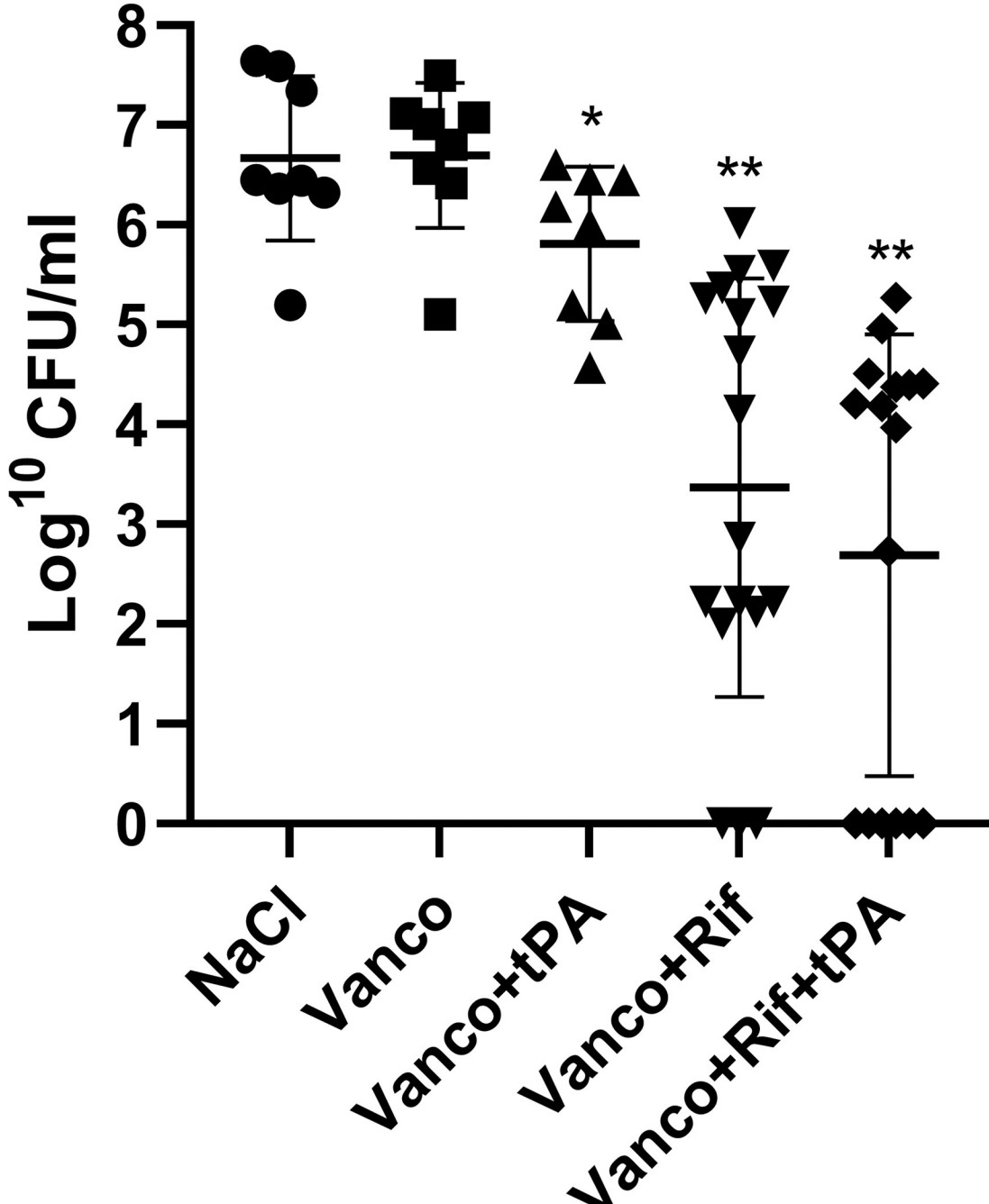

**Fig 6. Effect of antibiotics and tPA on biofilms formed on grafts *in vivo* after a 7-day treatment period.** Each data point represents median Log$^{10}$ CFU/mL of three replicates from one graft. Error bars represent median with IQR. Pairwise comparison of treatment groups (Mann-Whitney test, *p≤0.05 **p≤0.005).

We demonstrated that rifampicin could significantly reduce the MBEC of vancomycin to a level that is close to the MIC of 1.4 mg/L previously established for MRSA FPR3757 [29]. Previous studies have also found a positive effect of adding rifampicin to vancomycin treatment against MRSA biofilms. However none of these found the same dramatic decrease in MBEC. Our findings could be attributed to strain differences, as a previous study found that the

MBEC range for vancomycin was 1–64 mg/L and 0.13–32 mg/L for rifampicin when tested against 40 different MRSA strains [30]

We were unable to demonstrate any benefit of adding tPA to vancomycin + rifampicin treatment both *in vitro* and *in vivo* (Table 1 and Fig 6). While our initial *in vitro* data demonstrated the biofilm could be partly dispersed (Fig 7) this did not increase antibiotic

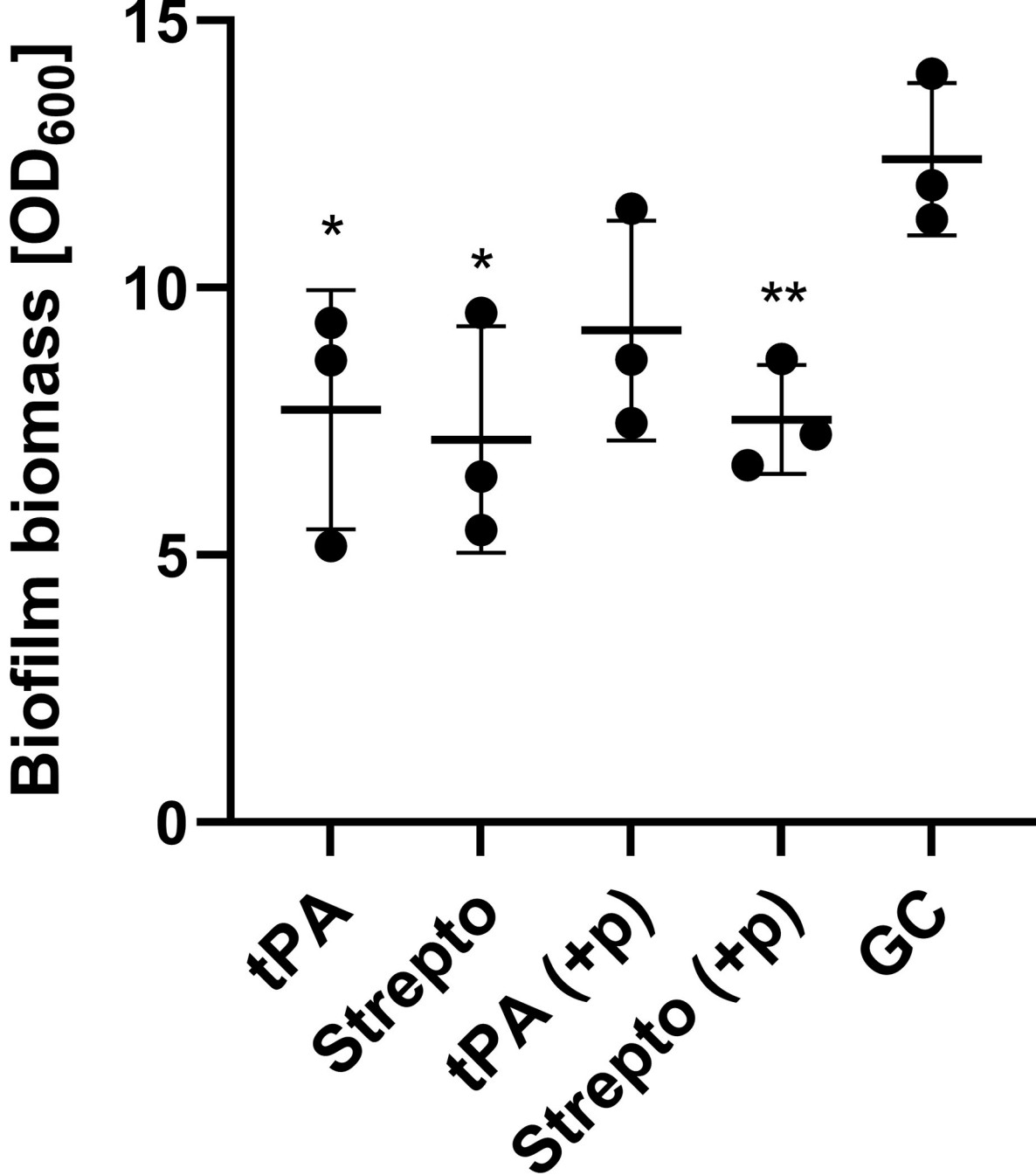

**Fig 7. Biofilm dispersal with tPA and streptokinase.** Mean biofilm biomass (± SD, n = 3) after 30 min treatment of 24 h old biofilm, measured by Gram's crystal violet staining. Each data point represents mean amount of biofilm biomass. tPA: tissue plasminogen activator 2500 U/ml, Strepto: Streptokinase 500 U/ml, GC: Growth control MRSA USA300 FPR3757, Blank: No bacteria, +p: Refers to plasma (10% heparin stabilized human plasma) added to treatment step. Pairwise comparison of treatment groups with growth control (Student's t-test, *P≤ 0.05 **P≤ 0.005).

susceptibility (Table 1). Contrary tPA seemed to have a protective effect on the biofilm against antibiotics and this could also explain why tPA was not able to increase antibiotic efficacy *in vivo*. These findings are in contrast with our previous findings on streptokinase's ability to improve efficacy of rifampicin-containing antibiotic therapy, but it must be noted that streptokinase had a much more dramatic reduction in biofilm biomass than observed in this study [13]. Enzymatic degradation of biofilm's matrix components in combination with antibiotics has previously shown promising results *in vivo* against device-related infections [11,15,31]. tPA used as implant coating have previously shown to increase antibiotic efficacy *in vivo* although tPA did not show any antimicrobial efficacy by itself [31]. In another study, Nattokinase and TrypLE showed a 100% cure rate in combination with rifampicin [15]. It should be noted that a control arm consisting of rifampicin monotherapy was not included, making it difficult to assess the impact of the different constituents of the interventions. Fibrinolysis by Nattokinase and TrypLE is not dependent on plasminogen as tPA is, and human tPA has shown less sensitivity on rat plasminogen *in vitro*, and therefore some studies have used a higher dose compared to this study [32]. A higher dose might have increased the antibiotic efficacy, but this would not be comparable to the clinical situation and the human dose has previously proven effective for thrombolysis in rats [19]. Finally, the rats had undergone vascular surgery just 10 days prior to starting tPA treatment and in the clinical setting, initiating tPA treatment at this point after major surgery is a relative contraindication [33]. We thusly opted for a more conservative dosage.

Our study also demonstrated that while vancomycin monotherapy was ineffective, combining vancomycin with rifampicin markedly improved treatment (Fig 6). In previous *in vivo S. aureus* biofilm models, vancomycin alone has shown varying effect on biofilm, with a decrease in mean log CFU between 0.2–1.6 [20,34,35]. The same studies showed that the addition of rifampicin yielded a cure rate of 71–100%. This is in contrast to the 16% cure rate we found, although there was a significant decrease in median bacterial load on grafts (3.50 vs. 6.56, p = 0.0016). While these results are from animal models on implant-associated osteomyelitis, there are clinical data supporting rifampicin combinations for PVGI [36]. Further studies are needed in order to clarify, which antibiotics benefit the most from rifampicin for acute PVGI caused by staphylococci. This is clinically relevant, given that the most recent guidelines from the European Society for Vascular Surgery on treatment of vascular graft infections focuses mainly on the use rifampicin as an applied or localized treatment option [18], rather than a systemically administered treatment.

Whilst this is the first model of a true PVGI in rats, the model has some limitations. Contrary to previous studies and the presumed avenue of graft infection in real-world clinical practice, we chose to inoculate the grafts prior to surgery. Previous studies have shown this method to yield high rates of infection and reduce biological variation in the burden of infection in each animal [37]. Another limitation is the choice of PE as the graft material and future studies should seek to use either PTFE or Dacron as the choice of material may affect graft patency, which we did not have the equipment to study. Graft patency has previously been studied with MRI scans in a similar murine model, where patency was preserved, although with a significant decrease in blood flow [28]. Finally, post-surgical heparinisation was omitted as heparin stimulates biofilm formation by *S. aureus* and this would impact the microbiological outcomes of the study [38,39].

Although animals tolerated both surgery and treatment well, animals receiving combination treatment had a slower growth rate (Fig 2). This could be attributed to a loss of appetite, a common side effect of rifampicin, or to nausea for the tPA-treated animals, caused by prolonged anaesthesia during iv injections. The heterogeneous sample size of our treatment groups is an obvious limitation to our study but was an intentional choice to comply the 3R

principles [40], whilst retaining the ability to test our hypothesis of whether tPA would improve treatment. This led to allocation of more animals to these two groups, as we felt it pertinent to test tPA against the most efficacious intervention.

Even under optimal conditions, not even half of our animals were cured from their vascular graft infection. Therefore a more potent antibiotic partner is needed, given the noted deficiency of vancomycin in clearing the infection. Future studies should investigate other conventional antibiotic treatment combinations with rifampicin and direct biofilm targeted therapy. This model is a valuable contribution to test hypotheses that could form the basis of future clinical studies.

## Conclusion

We developed the first functional model of an arterial prosthetic vascular graft infection in rats. Antibiotic combination therapy with vancomycin and rifampicin was superior to vancomycin monotherapy, and the addition of tPA did not significantly reduce bacterial load, nor significantly increase cure rate.

## Supporting information

**S1 Fig. Confocal laser scanning microscopy images of the MRSA biofilm formed in prosthetic vascular grafts in rats after 10 days of infection.** CLSM images of the biofilm formed at the vascular graft dissected crosswise (A) and lengthwise (B). Cells (bacterial and murine) are shown in blue (SYTO41) and fibrin is shown in green (anti-fibrin antibody, 59D8, Atto488 conjugated rabbit IgG). The 2D images are shown as two-channel. Scale bar 50 μm.
(PDF)

**S1 Table. Animal Research: Reporting of In Vivo Experiments (ARRIVE) guidelines checklist.**
(PDF)

## Acknowledgments

Parts of the study was presented at ISSSI 2018 conference in Copenhagen, Denmark, poster number P297

The authors wish to thank associate Professor Jacob Budtz-Lilly from the Department of Vascular Surgery, Aarhus University Hospital for excellent assistance in revision of this paper.

The Department of Clinical Microbiology at Aarhus University Hospital are thanked for its assistance in the microbiology laboratory.

## Author Contributions

**Conceptualization:** Søren Jensen-Fangel, Lars Østergaard, Rikke Louise Meyer, Nis Pedersen Jørgensen.

**Data curation:** Mikkel Illemann Johansen, Søren Jensen Rahbek, Gabriel Antonio S. Minero, Louise Kruse Jensen, Ole Halfdan Larsen, Lise Tornvig Erikstrup, Anders Marthinsen Seefeldt.

**Formal analysis:** Mikkel Illemann Johansen, Søren Jensen Rahbek, Gabriel Antonio S. Minero, Louise Kruse Jensen, Ole Halfdan Larsen, Lise Tornvig Erikstrup.

**Funding acquisition:** Mikkel Illemann Johansen, Nis Pedersen Jørgensen.

**Investigation:** Mikkel Illemann Johansen, Søren Jensen Rahbek, Gabriel Antonio S. Minero, Louise Kruse Jensen, Ole Halfdan Larsen, Lise Tornvig Erikstrup, Anders Marthinsen Seefeldt.

**Methodology:** Mikkel Illemann Johansen, Søren Jensen Rahbek, Rikke Louise Meyer, Nis Pedersen Jørgensen.

**Project administration:** Søren Jensen-Fangel, Lars Østergaard, Rikke Louise Meyer, Nis Pedersen Jørgensen.

**Resources:** Søren Jensen-Fangel, Gabriel Antonio S. Minero, Louise Kruse Jensen, Ole Halfdan Larsen, Lise Tornvig Erikstrup, Lars Østergaard, Rikke Louise Meyer.

**Supervision:** Søren Jensen-Fangel, Lars Østergaard, Rikke Louise Meyer, Nis Pedersen Jørgensen.

**Validation:** Mikkel Illemann Johansen, Rikke Louise Meyer, Nis Pedersen Jørgensen.

**Visualization:** Mikkel Illemann Johansen, Gabriel Antonio S. Minero, Louise Kruse Jensen.

**Writing – original draft:** Mikkel Illemann Johansen.

**Writing – review & editing:** Mikkel Illemann Johansen, Søren Jensen Rahbek, Søren Jensen-Fangel, Gabriel Antonio S. Minero, Louise Kruse Jensen, Ole Halfdan Larsen, Lise Tornvig Erikstrup, Anders Marthinsen Seefeldt, Lars Østergaard, Rikke Louise Meyer, Nis Pedersen Jørgensen.

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
