## [Decision Letter · Decision Letter 0]

18 Jan 2023

PONE-D-22-33479Fibrinolytic and antibiotic treatment of prosthetic vascular graft infections in a novel rat modelPLOS ONE

Dear Dr. Johansen,

Thank you for submitting your manuscript to PLOS ONE. After careful consideration, we feel that it has merit but does not fully meet PLOS ONE’s publication criteria as it currently stands. Therefore, we invite you to submit a revised version of the manuscript that addresses the points raised during the review process.

Of particular concern is both the statistical analysis/sample size for the different animal groups, as well as the validity of a fibrinous biofilm. Within that context, the lack of an effect of TPA, in vivo, raised some concerns about the nature of the biofilm, and perhaps even the dosing regiment that was used in your model.

We look forward to receiving your revised manuscript.

Kind regards,

Noreen J. Hickok, Ph.D.

Academic Editor

PLOS ONE

Journal Requirements:

"This work was supported by Aarhus University Research Foundation and Knud og Edith Eriksen Mindefond."

"This work was supported by a PhD research grant from the Graduate School of Health, Aarhus University (MIJ), by Aarhus University Research Foundation (https://auff.au.dk/en/) (MIJ) and Knud og Edith Eriksen Mindefond (Grant number 62786) (NPJ)(https://www.keemindefond.dk/). The funders did not play any role in the study design, data collection and analysis, decision to publish or preparation of the manuscript."

Reviewers' comments:

Reviewer's Responses to Questions

**Comments to the Author**

1. Is the manuscript technically sound, and do the data support the conclusions?

Reviewer #1: Partly

Reviewer #2: Partly

2. Has the statistical analysis been performed appropriately and rigorously? 

Reviewer #1: I Don't Know

Reviewer #2: Yes

3. Have the authors made all data underlying the findings in their manuscript fully available?

Reviewer #1: Yes

Reviewer #2: No

4. Is the manuscript presented in an intelligible fashion and written in standard English?

Reviewer #1: Yes

Reviewer #2: Yes

5. Review Comments to the Author

Reviewer #1: Material/implant associated infections are still feared complications after surgery and can have devastating consequences for the patient. Infections of the vascular graft cause morbidity and can lead to death. Improved treatment options of the infection and the biofilm are therefore urgently needed. The authors of the study present a new rat model to investigate prosthetic vascular graft infection and possible treatment options. The model resulted in an infection rate 100% with survival of the animals over the experimental period. They tested the efficacy of tissue plasminogen activator (tPA) in vitro and showed a biofilm reduction of about 30%. In vitro evaluation of the effect of tPA plus antibiotics was not done. They went into the animal model and found a significant but only minor reduction of the CFU after antibiotic treatment, addition of tPA had no effect.

1. Line 56 ff. The cited studies are no all in vivo studies on IVC infection. Please be more careful with the citation of studies.

2. Line 70 ff: The authors explain the sample size calculation. I am not an expert in statistics, but it seems unusual to calculate the sample size only for the treatment group. They used a sample size of n=4 for the control and up to n=18 for the treatment groups. This difference needs further explanation.

3. Due to the comparison of multiple groups, a post hoc test is necessary.

4. Line 184 ff: please provide the values for the control group. Please give the unit for the values.

5. Fig. 2: Please show the individual data points (dot plots as in the other graphs)

6. The effect of tPA is not very pronounced compared to other substances and studies (see Ref 16) and it should be discussed why they used this substance with the low efficacy.

7. In a previous study (Microorganisms 2016, 4, 36, Ref 13) they saw a much higher biofilm dispersal with lower Steptokinase concentrations (500 vs 2500 U/ml). Please discuss this.

8. Why was not the combination of tPA plus antibiotics tested in vitro? This would have provided valuable data regarding the possible effect and might have foreseen the failure of the animal study.

9. Table 1. Please add the information also for the other groups.

10. Fig. 4: not all animals showed a systemic infection. However, in the abstract they mention infection in all animals. Please clarify.

11. Figure legends: Please check the sample size for the vanco-group. It should be n=5 as shown in Fig. 5 and described in the drop-out section.

12. They see a reduction from 6.56 log10 CFU/mL (Control) to 3.50 log10 CFU/mL (V+R). This effect is statistically significant, but is it clinically meaningful? Normally a log3 reduction is expected to show antimicrobial efficacy. This should be discussed.

13. They discuss several limitations and explain the rational for the study design. However, they do not address the limited in vitro effect, the missing antibiotics group in the in vitro study and why they went with this effect to the in vivo model.

14. Please check the separation of decimals within the text.

Reviewer #2: General comments

This is generally well written and potential valuable paper. However, I have two main concerns. I understand that the authors justifiably strived to use as few animals as possible, but I feel an important control is missing - to assess the impact and integration of the procedure alone by using an uninoculated graft. In addition to weight change the graft and surrounding tissue could be assessed after 10 and 18 days (i.e. immediately prior to treatment and at termination) by H&E or visual inspection. Another criticism is that although the animal model is designed to be a biofilm infection model there is no direct demonstration of biofilm on the untreated control graft. A further control would be to assess for biofilm at days 10 and at 18. If a fibrin coated biofilm did not form might this explain the limited activity of tPA? Confocal or SEM demonstrating biofilm in controls as compared to treated animals would significantly strengthen the manuscript. The grafts could have been sectioned for one piece to be imaged and the other for plate count.

Specific comments:

1) It would be useful to provide vancomycin MIC and also to assess, or provide a reference, on whether vanco and rifamp act synergistically or additively.

2) Why was tPA chosen for the animal studies over streptokinase?

3) It would be useful to show (or reference) whether tPA or streptokinase have any kind of antimicrobial activity.

4) can the authors explain why the untreated rats had a higher CFU load on the graft but also showed the most weight gain?

5) Fig. 5. The vanc + rifamp combinations show bimodal distributions. Is this possibly an artefact of plotting 0 CFU as zero rather than as the detection limit? If the bimodal distribution is real is there any evidence of why those particular mice may have cleared the infection from post-mortem exam during dissection?

6. PLOS authors have the option to publish the peer review history of their article (what does this mean?). If published, this will include your full peer review and any attached files.

Reviewer #1: No

Reviewer #2: No

---

## [Author Response · Author response to Decision Letter 0]

22 May 2023

Reviewer #1: Material/implant associated infections are still feared complications after surgery and can have devastating consequences for the patient. Infections of the vascular graft cause morbidity and can lead to death. Improved treatment options of the infection and the biofilm are therefore urgently needed. The authors of the study present a new rat model to investigate prosthetic vascular graft infection and possible treatment options. The model resulted in an infection rate 100% with survival of the animals over the experimental period. They tested the efficacy of tissue plasminogen activator (tPA) in vitro and showed a biofilm reduction of about 30%. In vitro evaluation of the effect of tPA plus antibiotics was not done. They went into the animal model and found a significant but only minor reduction of the CFU after antibiotic treatment, addition of tPA had no effect.

1. Line 56 ff. The cited studies are no all in vivo studies on IVC infection. Please be more careful with the citation of studies.

We thank the reviewer for making us aware of this mistake, this has been changed in the manuscript line 64.

2. Line 70 ff: The authors explain the sample size calculation. I am not an expert in statistics, but it seems unusual to calculate the sample size only for the treatment group. They used a sample size of n=4 for the control and up to n=18 for the treatment groups. This difference needs further explanation.

We agree that this is an unusual way of calculating sample size. Previous in vivo studies on antibiotic treatment of osteomyelitis in rats have also used diverging sample sizes in the different treatment arms [1, 2]. These studies did not include a sample size calculation, as we did. We initially did a pilot study with n=6 in each treatment arm to find the best antibiotic treatment to test tPA together with, since this was our main aim. Vancomycin and Rifampicin had the highest reduction in bacterial load on the grafts, and therefore we chose to calculate the final sample size based on a significant difference between vancomycin+rifampicin+tPA and vancomycin+rifampicin. We do however agree that the sample size of the NaCl group should have been larger to compare it to the other groups. Therefore, we have now included additional animals to increase the sample size of the NaCl, vancomycin and vancomycin+tPA group to n=8 for each group, so that they had an equal sample size (revised fig 7). 

3. Due to the comparison of multiple groups, a post hoc test is necessary. 

This is an important observation, which we agree on. All p-values have now been tested for type-1 errors using the Holm-Bonferroni test. The post hoc analysis did not lead to changes in our conclusions. 

4. Line 184 ff: please provide the values for the control group. Please give the unit for the values. 

Thank you for this comment, we agree that the wording could have been clearer, and therefore we have now revised the result section of the biofilm dispersal study (line 256-261). We chose to add the amount of reduction of each treatment when compared to the growth control, so that the effect of each treatment is more apparent. The values are the mean biofilm biomass measured as optical density.

5. Fig. 2: Please show the individual data points (dot plots as in the other graphs)

Fig 2 has now been revised to a dot plot 

6. The effect of tPA is not very pronounced compared to other substances and studies (see Ref 16) and it should be discussed why they used this substance with the low efficacy.

We thank the reviewer for this comment. The reason for choosing tPA is that it is an FDA approved drug already in wide use for stroke patients. As such we thought it ideal for drug repurposing. This is opposed to TrypLE and nattokinase which are not FDA approved drugs. TrypLE and Nattokinase have been tested in vivo as catheter lock solutions where the drugs are installed as a local treatment at the site of infection however, we do not know if the antibiofilm effect of these drug would be as pronounced when given as iv injection or what adverse events such therapy would have [3]. Furthermore streptokinase works on free plasminogen resulting in fibrinolysis in the whole body which can cause thrombosis and hemorrhage and have shown to increase mortality compared to placebo [4]. TrypLE and nattokinase are both serine proteases that work directly on fibrin and we therefore considered in our design of this study that there would be risks involved using these drugs. tPA on the other hand only works on fibrin bound plasminogen and therefore it only has an effect around a thrombus or a fibrin covered biofilm. Therefore, we chose tPA as we felt it the most clinically relevant of the mentioned drugs.

7. In a previous study (Microorganisms 2016, 4, 36, Ref 13) they saw a much higher biofilm dispersal with lower Steptokinase concentrations (500 vs 2500 U/ml). Please discuss this. 

Thank you for making us aware of this. We did actually use 500 U/mL streptokinase in this study, and the 2500 U/mL written in the manuscript is a type writing mistake, we apologize for that. We agree that it is a less pronounced effect of streptokinase, than seen previously, which we also touch upon in the manuscript line 424-428. The reason for this could be that we only used 10% plasma in our biofilms instead of 50%, and therefore there were less fibrin in the biofilm for the streptokinase and tPA to work on. We chose to reduce the plasma amount since 50% plasma with this isolate resulted in very dense biofilms in our initial experiments which were difficult to use in in vitro studies. Another reason for the difference in biofilm dispersal could be that we used another S. aureus strain than we used in Jørgensen et al [5]. Biofilm formation is very strain dependent and our S. aureus strain may incorporate less fibrin in its extracellular matrix. 

8. Why was not the combination of tPA plus antibiotics tested in vitro? 

This would have provided valuable data regarding the possible effect and might have foreseen the failure of the animal study. Based on this comment, we have performed an additional in vitro experiment to test if vancomycin MBEC was lowered +/- tPA and +/- rifampicin. This has been added to line 109-118 and line 274-282. We initially abstained from this, as we felt that the experimental conditions in the in vitro setup and the in vivo model were not comparable, and a negative finding in vitro not necessarily would rule out effect in vivo, as we would be delivering multiple doses into a living animal with a functioning coagulation system as well as an immune system. Although we did not find any effect of adding tPA to antibiotic treatment in vitro, there can be vast differences between an in vitro and an in vivo grown biofilm, among others gene expression and matrix composition, which can affect antibiotic tolerance [6]. Therefore it is important to test new treatment against complicated infections – such as PVGI, in different experimental setups. 

9. Table 1. Please add the information also for the other groups. 

Table 1 (changed to table 2 in the manuscript) was shown to demonstrate whether rifampicin resistance had developed in animals treated with rifampicin only. Since emergence of vancomycin resistance in Staphylococcus aureus during therapy is very rarely seen both in vivo and in patients, we chose to only present the groups that were treated with rifampicin. However, we have now added amount of plates with and without growth from the other treatment groups to table 2. 

10. Fig. 4: not all animals showed a systemic infection. However, in the abstract they mention infection in all animals. Please clarify. 

We have changed the wording in the abstract in line 32-35 and the wording in the result section line 406-407, to be more accurate. However all animals in the control group (NaCl) had an average log 3 CFU/ml increase in bacterial load on the grafts from implantation and until it was removed and we observed very high bacterial loads in all animals in the control group at the end of the study. This, to us, proves that the animals were infected, and that we developed a PVGI model with a high infectious load that was not cleared by the animal itself. Figure 4 demonstrates the rate at which disseminated infection was identified. Why this only occurred in 60% of untreated animals remains unanswered. Legout et al found only 29 positive blood cultures in a cohort of 85 confirmed PVGI cases in a prospective observational cohort study [7]. The infection rate in the untreated group is 100% when the grafts are investigated (Fig 7). 

11. Figure legends: Please check the sample size for the vanco-group. It should be n=5 as shown in Fig. 5 and described in the drop-out section. 

Thank you for making us aware of this, this has been changed to n=8 after we increased the sample size of this group. 

12. They see a reduction from 6.56 log10 CFU/mL (Control) to 3.50 log10 CFU/mL (V+R). This effect is statistically significant, but is it clinically meaningful? Normally a log3 reduction is expected to show antimicrobial efficacy. This should be discussed. 

We agree that it is always important to question the translational aspect of experimental studies. Whether the log reduction observed in this study would lead to a succesfull treatment outcome clinically is difficult to answer due to the following aspects. Firstly, at which bacterial load is the host immune system able to clear the infection? Secondly, the treatment duration in this study is much shorter than the typically used treatment duration for PVGI (>6weeks) – what would have happened, had the treatment been prolonged? Thirdly, what constitutes a robust treatment response in an animal model? The first point is difficult to address for PVGI models as such an experiment has to the best of our knowledge not been performed. To the second, a previous study in a rat implant-associated osteomyelitis model have shown a decreased effect of V+R treatment for 21 days of treatment compared to 48 h treatment [8]. However, there was a trend towards that treatment with V+R was still effective after 2 weeks of treatment although the median CFU on implants was 2 log higher compared to 48 h of treatment [8]. It should also be added that the log10 CFU/mL in our study is the median value and some of the animals had a bacterial load on the grafts that were below the detection limit. Finally, implant-associated animal models typically report good treatment outcomes with 4-5 reduction in log CFU on implants [2, 9, 10] Based on the reviewers input, we have amended line 390-391, and line 442-443 in the manuscript. 

13. They discuss several limitations and explain the rational for the study design. However, they do not address the limited in vitro effect, the missing antibiotics group in the in vitro study and why they went with this effect to the in vivo model. 

As the reviewer mentions there are several limitations in our study, which we have adressed. In the answer to comment 7, we have addressed the issue of our lower in vitro effect of tPA and streptokinase compared to other studies. The reason for our choice of fibrinolytic is discussed in comment 6. In regards to the missing antibiotic group we agree with the reviewer that this should have been done initially, and therefore we have now done MBEC assays of our antibiotics with and without tPA, so that all drugs in the study have been investigated in vitro and in vivo (line 109-118 and line 274-282). Including other antibiotics or enzymes is beyond the scope of this work. 

14. Please check the separation of decimals within the text. 

This has now been checked and revised in the manuscript 

Reviewer #2: General comments This is generally well written and potential valuable paper. However, I have two main concerns. I understand that the authors justifiably strived to use as few animals as possible, but I feel an important control is missing - to assess the impact and integration of the procedure alone by using an uninoculated graft. In addition to weight change the graft and surrounding tissue could be assessed after 10 and 18 days (i.e. immediately prior to treatment and at termination) by H&E or visual inspection. Another criticism is that although the animal model is designed to be a biofilm infection model there is no direct demonstration of biofilm on the untreated control graft. A further control would be to assess for biofilm at days 10 and at 18. If a fibrin coated biofilm did not form might this explain the limited activity of tPA? Confocal or SEM demonstrating biofilm in controls as compared to treated animals would significantly strengthen the manuscript. The grafts could have been sectioned for one piece to be imaged and the other for plate count. 

We thank the reviewer for the kind appraisal and for pointing out these important issues. We agree that the addition of a control assessing the impact and integration of the surgical procedure is necessary. To satisfy these points we have conducted a range of additional experiments.

Firstly, we have included 6 animals for histological analysis of the graft and surrounding tissue 19 days after surgery (see line 211-242 for methods and line 337-388 for results in the manuscript). Further, we have included an additional 9 animals for Confocal Laser Scanning Microscopy of grafts after 10 and 19 days of infection to demonstrate that there was a fibrin containing biofilm on the grafts. The methods applied and results of the CLSM studies can be seen at line 196-209 for methods and line 318-329 for results, in the manuscript. 

Specific comments:

1) It would be useful to provide vancomycin MIC and also to assess, or provide a reference, on whether vanco and rifamp act synergistically or additively. 

Thank you for this comment. The MIC for vancomycin has already been established for MRSA USA300 fpr3757 to 1.4 ug/ml [11]. We understand the scientific value of adding this to the paper however, our aim was to test antibiotics +/- tPA against biofilm infections and not planktonic bacteria. Therefore we think an MBEC assay would be more appropriate, which we have now added to the manuscript, to test antibiotics +/- tPA in vitro (line 109-118 and line 274-282). To test synergism we would need to do a checkerboard assay. We agree that this would be interesting but it is not something you typically do in in vivo studies and therefore we feel that this is beyond scope of the study. 

2) Why was tPA chosen for the animal studies over streptokinase? 

An important question that is raised which we did address prior in response to comment 6 from reviewer 1 and have amended the manuscript to further clarify this in line 59-63. We wanted to use clinically approved drugs as to ensure that our study was translational and easily implementable in the clinic. Streptokinase has been outphased from the Danish hospitals, because of an increased risk of hemorrhage, and been replaced with tPA. Since patients suffering from acute PVGI have recently undergone a major surgery, we did not find it safe to use a drug with an increased risk of hemorrhage. Therefore, we chose tPA although it had a less pronounced biofilm dispersing effect than streptokinase. 

3) It would be useful to show (or reference) whether tPA or streptokinase have any kind of antimicrobial activity. 

A previous study has demonstrated that tPA has no antimicrobial effect using a polystyrene coverslip which was coated with tPA prior to subcutaneous insertion in to mice. This has been added to the discussion section line 430-432 [12].

4) can the authors explain why the untreated rats had a higher CFU load on the graft but also showed the most weight gain? 

Since the initial weight loss is approximately equivalent in all treatment groups, we presume that the reduced weight gain in the other treatment groups could be attributed to the antimicrobial treatment and prolonged anesthesia during administration. This is explained in more detail in the manuscript line 462-464. 

5) Fig. 5. The vanc + rifamp combinations show bimodal distributions. Is this possibly an artefact of plotting 0 CFU as zero rather than as the detection limit? If the bimodal distribution is real is there any evidence of why those particular mice may have cleared the infection from post-mortem exam during dissection?

We thank the reviewer for this important observation. Following CFU measurements of sonicate from explanted grafts, the remaining sonicate and the graft from each animal were added to 20 mL of fresh BHI media and grown overnight at 37°C. This was done to increase the bacterial detection limit on both graft and organs and have previously been done by both Jørgensen et al and Metsemakers et al [13, 14]. This has been added to the methods section line 178-182. All animals plotted as 0 CFU/mL had no growth on the agar plate or in the overnight culture of the graft with remaining sonicate. We feel confident that these animals have indeed cleared the infection in the prosthesis. It was however not possible to definitively assess macroscopically if the grafts were infected when inspecting them with the naked eye.

References

1. Albac S, Labrousse D, Hayez D, Anzala N, Bonnot D, Chavanet P, et al. Activity of Different Antistaphylococcal Therapies, Alone or Combined, in a Rat Model of Methicillin-Resistant Staphylococcus epidermidis Osteitis without Implant. Antimicrob Agents Chemother. 2020;64(2).

2. Park KH, Greenwood-Quaintance KE, Mandrekar J, Patel R. Activity of Tedizolid in Methicillin-Resistant Staphylococcus aureus Experimental Foreign Body-Associated Osteomyelitis. Antimicrob Agents Chemother. 2016;60(11):6568-72.

3. Hogan S, O'Gara JP, O'Neill E. Novel Treatment of Staphylococcus aureus Device-Related Infections Using Fibrinolytic Agents. Antimicrob Agents Chemother. 2018;62(2).

4. Bivard A, Lin L, Parsonsb MW. Review of stroke thrombolytics. J Stroke. 2013;15(2):90-8.

5. Jørgensen NP, Zobek N, Dreier C, Haaber J, Ingmer H, Larsen OH, et al. Streptokinase Treatment Reverses Biofilm-Associated Antibiotic Resistance in Staphylococcus aureus. Microorganisms. 2016;4(3).

6. Coenye T, Nelis HJ. In vitro and in vivo model systems to study microbial biofilm formation. J Microbiol Methods. 2010;83(2):89-105.

7. Legout L, Sarraz-Bournet B, D'Elia PV, Devos P, Pasquet A, Caillaux M, et al. Characteristics and prognosis in patients with prosthetic vascular graft infection: a prospective observational cohort study. Clin Microbiol Infect. 2012;18(4):352-8.

8. Vergidis P, Schmidt-Malan SM, Mandrekar JN, Steckelberg JM, Patel R. Comparative activities of vancomycin, tigecycline and rifampin in a rat model of methicillin-resistant Staphylococcus aureus osteomyelitis. J Infect. 2015;70(6):609-15.

9. Niska JA, Shahbazian JH, Ramos RI, Francis KP, Bernthal NM, Miller LS. Vancomycin-rifampin combination therapy has enhanced efficacy against an experimental Staphylococcus aureus prosthetic joint infection. Antimicrob Agents Chemother. 2013;57(10):5080-6.

10. Saleh-Mghir A, Muller-Serieys C, Dinh A, Massias L, Crémieux AC. Adjunctive rifampin is crucial to optimizing daptomycin efficacy against rabbit prosthetic joint infection due to methicillin-resistant Staphylococcus aureus. Antimicrob Agents Chemother. 2011;55(10):4589-93.

11. Friberg C, Haaber JK, Vestergaard M, Fait A, Perrot V, Levin BR, et al. Human antimicrobial peptide, LL-37, induces non-inheritable reduced susceptibility to vancomycin in Staphylococcus aureus. Sci Rep. 2020;10(1):13121.

12. Kwiecinski J, Na M, Jarneborn A, Jacobsson G, Peetermans M, Verhamme P, et al. Tissue Plasminogen Activator Coating on Implant Surfaces Reduces Staphylococcus aureus Biofilm Formation. Appl Environ Microbiol. 2016;82(1):394-401.

13. Jørgensen NP, Hansen K, Andreasen CM, Pedersen M, Fuursted K, Meyer RL, et al. Hyperbaric Oxygen Therapy is Ineffective as an Adjuvant to Daptomycin with Rifampicin Treatment in a Murine Model of Staphylococcus aureus in Implant-Associated Osteomyelitis. Microorganisms. 2017;5(2).

14. Metsemakers WJ, Emanuel N, Cohen O, Reichart M, Potapova I, Schmid T, et al. A doxycycline-loaded polymer-lipid encapsulation matrix coating for the prevention of implant-related osteomyelitis due to doxycycline-resistant methicillin-resistant Staphylococcus aureus. J Control Release. 2015;209:47-56.

---

## [Decision Letter · Decision Letter 1]

12 Jun 2023

Fibrinolytic and antibiotic treatment of prosthetic vascular graft infections in a novel rat model

PONE-D-22-33479R1

Dear Dr. Johansen,

We’re pleased to inform you that your manuscript has been judged scientifically suitable for publication and will be formally accepted for publication once it meets all outstanding technical requirements.

Kind regards,

Noreen J. Hickok, Ph.D.

Academic Editor

PLOS ONE

Additional Editor Comments (optional):

Reviewers' comments:

Reviewer's Responses to Questions

**Comments to the Author**

1. If the authors have adequately addressed your comments raised in a previous round of review and you feel that this manuscript is now acceptable for publication, you may indicate that here to bypass the “Comments to the Author” section, enter your conflict of interest statement in the “Confidential to Editor” section, and submit your "Accept" recommendation.

Reviewer #1: All comments have been addressed

Reviewer #2: All comments have been addressed

2. Is the manuscript technically sound, and do the data support the conclusions?

Reviewer #1: Yes

Reviewer #2: Yes

3. Has the statistical analysis been performed appropriately and rigorously? 

Reviewer #1: Yes

Reviewer #2: Yes

4. Have the authors made all data underlying the findings in their manuscript fully available?

Reviewer #1: Yes

Reviewer #2: (No Response)

5. Is the manuscript presented in an intelligible fashion and written in standard English?

Reviewer #1: Yes

Reviewer #2: Yes

6. Review Comments to the Author

Reviewer #1: I highly appreciate the additional work of the authors. They increased the number of animals in selected groups and added further in vitro and in vivo data. All my comments are satisfactorily answered. Even though the new in vitro experiment do not highly support the in vivo study, the argumentation of the authors is clear and I agree.

Please check/correct the amount of Streptokinase used in the Biofilm assay (M&M): 500 instead of 2500 U/ml as mentioned in the rebuttal letter and in the figure legend.

Reviewer #2: The authors have done a good job of doing additional experiments and addressing my concerns. This work will be a valuable contribution to the field.

7. PLOS authors have the option to publish the peer review history of their article (what does this mean?). If published, this will include your full peer review and any attached files.

Reviewer #1: No

Reviewer #2: **Yes: **Paul Stoodley

---

## [Editor Report · Acceptance letter]

10 Jul 2023

PONE-D-22-33479R1 

Fibrinolytic and antibiotic treatment of prosthetic vascular graft infections in a novel rat model 

Dear Dr. Johansen:

I'm pleased to inform you that your manuscript has been deemed suitable for publication in PLOS ONE. Congratulations! Your manuscript is now with our production department. 

Kind regards, 

on behalf of

Dr. Noreen J. Hickok 

Academic Editor

PLOS ONE